# Seed Priming with Glass Waste Microparticles and Red Light Irradiation Mitigates Thermal and Water Stresses in Seedlings of *Moringa oleifera*

**DOI:** 10.3390/plants11192510

**Published:** 2022-09-26

**Authors:** Patrícia da Silva Costa, Rener Luciano de Souza Ferraz, José Dantas Neto, Semako Ibrahim Bonou, Igor Eneas Cavalcante, Rayanne Silva de Alencar, Yuri Lima Melo, Ivomberg Dourado Magalhães, Ashwell Rungano Ndhlala, Ricardo Schneider, Carlos Alberto Vieira de Azevedo, Alberto Soares de Melo

**Affiliations:** 1Academic Unit of Agricultural Engineering, Federal University of Campina Grande, Campina Grande 58428-830, Paraíba, Brazil; 2Academic Unit of Development Technology, Federal University of Campina Grande, Sumé 58540-000, Paraíba, Brazil; 3Department of Plant Science and Environmental Sciences, Federal University of Paraíba, Areia 58051-900, Paraíba, Brazil; 4Department of Biology, State University of Paraíba, Campina Grande 58429-500, Paraíba, Brazil; 5Department of Agronomy, Federal University of Alagoas, Rio Largo 57100-000, Alagoas, Brazil; 6Green Biotechnologies Research Centre of Excellence, University of Limpopo, Sovenga 0727, Limpopo, South Africa; 7Department of Chemistry, Federal Technological University of Paraná, Toledo 85902-000, Paraná, Brazil

**Keywords:** Moringaceae, abiotic stresses, gas exchange, cell membrane integrity, water status, osmotic adjustment, antioxidant mechanism

## Abstract

The association between population increase and the exploitation of natural resources and climate change influences the demand for food, especially in semi-arid regions, highlighting the need for technologies that could provide cultivated species with better adaptation to agroecosystems. Additionally, developing cultivation technologies that employ waste materials is highly desirable for sustainable development. From this perspective, this study aimed to evaluate whether seed priming with glass waste microparticles used as a silicon source under red light irradiation mitigates the effects of thermal and water stress on seedlings of *Moringa oleifera*. The experimental design was set up in randomized blocks using a 2 × 2 × 2 factorial arrangement consisting of seed priming (NSP—no seed priming, and SPSi—seed priming with glass microparticles under red light irradiation), soil water replenishment (W50—50%, and W100—100% of crop evapotranspiration—ETc), and temperature change (TC30°—30 °C day/25 °C night and TC40°—40 °C day/35 °C night). Seed priming with glass microparticles under red light irradiation mitigated the effects of thermal and water stress on seedlings of *Moringa oleifera* seedlings through the homeostasis of gas exchange, leaf water status, osmotic adjustment, and the antioxidant mechanism.

## 1. Introduction

The steady growth of the world population presents the agricultural sector with a challenge to increase food production and ensure food security, especially because the number of undernourished people worldwide grew from 83 to 132 million in 2020. However, the increase in production has to occur with the lowest possible impact on natural resources in order to meet one of the priority goals of the 2030 Agenda for Sustainable Development [1]. From this perspective, one alternative to increase food production and reduce the impacts of agriculture on agroecosystems is growing plants with the potential for multiple uses and genotypic and phenotypic plasticity for different cultivation environments [2].

In this scenario, *Moringa oleifera* Lamarck, a species of the family Moringaceae and native to India and Pakistan [3], could be one such alternative for cultivation given its various food, medicinal, industrial, environmental, and social purposes, and as an abundant source of essential amino acids, macronutrients, and micronutrients. This species also shows outstanding anti-fungal, analgesic, anti-inflammatory, anti-oxidant, anti-diabetic, anti-tumoral, and anti-bacterial properties, with high seed oil contents [4,5,6,7,8]. Furthermore, the species can also be used to purify water, generate income, and improve the quality of life of producers around the world [4,5,6,7,8].

However, despite the adaptive potential of *M. oleifera* to agroecosystems, its development, growth, and production can decrease when the plant is exposed to thermal and water stresses [9,10,11,12], especially in semi-arid regions with high solar radiation levels, increased air temperature, and soil water restriction, conditions that could be further aggravated by environmental climate changes [13,14]. Thermal stress reduces seed germination and vigor in seedlings of *M. oleifera* [15,16] and increases the accumulation of non-structural carbohydrates, amino acids, and phenols [12]. Water stress activates the antioxidant mechanism of this species [10], which, in turn, reduces the stomatal size, the content of photosynthetic pigments, leaf gas exchange, and root and shoot growth [9,11,17].

This scenario highlights the desire for technologies that can mitigate the effects of abiotic stresses on *M. oleifera*. In that regard, seed priming emerges as a promising alternative [2] that consists of soaking seeds in stress-mitigating solutions, e.g., silicon microparticle sources (SiMPs), with verified positive results in stress management in other crops, because the silicon is deposited in the endoplasmic reticulum, cell walls and intercellular spaces as hydrated amorphous silica, in addition to forming complexes with polyphenols to reinforce cell walls [18,19,20,21]; for example, wheat (*Triticum aestivum* L.) under cadmium (Cd) stress [22,23] and thermal stress [24], and rice (*Oryza sativa* L.) under drought [25]. Furthermore, developing accessible methodologies to obtain materials that will serve as Si sources is highly desirable. For example, the application of glass waste, a material easily found in landfill sites and used as silicon microparticle powders (SiMPs), becomes a simple approach that only requires grinding and sieving glass waste. Therefore, considering that seeds of *M. oleifera* are responsive to light [16], especially monochromatic red light [2], the irradiation of this type of light during seed priming could potentialize the stress-mitigating effect of SiMPs on plants.

From this perspective, we hypothesized that the seedlings of *M. oleifera* generated by seed priming with residual glass microparticles used as a Si source, or SiMPs, and under irradiation with monochromatic red light (RL), changed their gas exchange, osmotic adjustment, antioxidant mechanism, and dry matter accumulation as responses to overcome the effects of abiotic stresses.

Therefore, this study aimed to evaluate whether seed priming with silicon microparticles and red-light irradiation mitigate the effects of thermal and water stress on seedlings of *Moringa oleifera* cultivated in a Phytotron growth chamber.

## 2. Results

### 2.1. Principal Components and Multivariate Variance

Three principal components (PCs) with eigenvalues (λ) higher than one and percentages of variance (s^2^) higher than 10% were formed by the linear combination between 13 original variables collected from seedlings of *M. oleifera* generated by seed priming (SP) with glass microparticles (SiMPs) and subjected to different combinations of soil water replenishment levels (SWR) and temperature variations (TC). The first three PCs explained 86.87% of s^2^. PC_1_ represented 53.67% of s^2^ and was formed by combining the leaf water status (relative water content—RWC), leaf gas exchange (stomatal conductance—gs, transpiration—E, and internal CO_2_ concentration—Ci), leaflet osmotic adjustment indicators (proline in leaves—PRO, total soluble proteins in leaves—TSP-L, and total soluble sugars in leaves—TSS-L) catalase activity in the leaflets (CAT-L), and total dry matter accumulation (TDM). PC_2_ represented 18.72% of s^2^ and was formed by the combination between the net photosynthetic rate (A), superoxide dismutase activity in the leaflets (SOD-L), and TDM accumulation. PC_3_ represented 14.49% of s^2^ and was formed by the osmotic adjustment indicator in the roots (total soluble proteins in roots—TSP-R) and the activity of the antioxidant mechanism of roots (CAT-R). Electrolyte leakage in the leaflets (EL), the total sugar content in roots (TSS-R), and SOD-R activity were not associated with any PC, and their s^2^ values were disregarded in the principal component analysis (PCA) to be subjected to univariate analysis of variance (ANOVA). There was a significant interaction (*p*-value < 0.01) between SP x SWR x TC in the three PCs, according to the results of the multivariate analysis of variance (MANOVA) (Table 1).

### 2.2. Responses to Thermal and Water Stresses and Mitigation by Seed Priming

In the two-dimensional projection of the first four PCs (Figure 1A,D), PC_1_ is seen as a process triggered by temperature variations (TC), PC_2_ is triggered by the soil water replenishment levels (SWR), and PC_3_ is triggered by seed priming (SP). In PC_1_, temperature variation with thermal stress (40 °C day/35 °C night) increased the RWC, gs, E, Ci, and CAT-L and reduced the leaflet osmotic adjustment (PRO, TSP-L, and TSS-L) and the total dry matter accumulation (TDM) in relation to the seedlings that were not subjected to water stress (30 °C day/25 °C night). However, when seed priming (SP) with glass microparticles used as a silicon source was applied under monochromatic red light (RL), the seedlings that were subjected to thermal stress (SPSi-W100-T40°) and water stress (SPSi-W50-T30°), either in isolation or combined (SPSi-W50-T40°), showed mitigation in such stresses by reducing the RWC, gs, E, Ci, and CAT-L and increasing the PRO, TSP-L, TSS-L, and TDM. On the other hand, in the seedlings that were not subjected to the previously mentioned stresses (SPSi-W100-T30°), SP increased the RWC, gs, E, Ci, and CAT-L and decreased the PRO, TSP-L, TSS-L, and TDM in relation to those that did not receive seed priming (NSP-W100-T30°) (Figure 1A,B).

In PC_2_, the water stress caused by 50% soil water replenishment (W50) increased the activity of the SOD-L enzyme and decreased the net photosynthetic rate (A) and TDM accumulation in the seedlings in relation to those that were not subjected to this condition (W100). SP with SiMPs under RL irradiation mitigated the combined effects of water and thermal stresses (SPSi-W50-T40°) in relation to stressed seedlings that were not generated by SP (NSP-W50-T40°) by inducing a lower SOD-L activity and higher A and TDM values. SP did not mitigate water stress when this condition was imposed in isolation (SPSi-W50-T30°) compared to seedlings produced from seeds that did not undergo SP (NSP-W50-T30°). It is also seen that, in seedlings that were not subjected to stresses (SPSi-W100-T30°), SP reduced the activity of SOD-L and increased the A and TDM values in relation to control seedlings (NSP-W100-T30°) (Figure 1A,B).

In PC_3_, SP application with SiMPs and RL irradiation increased the total soluble protein content in the roots (TSP-R) and reduced catalase activity in the roots (CAT-R) of seedlings subjected to thermal (SPSi-W100-T40°) and water stress (SPSi-W50-T30°) in isolation and also when these stresses were simultaneously imposed (SPSi-W50-T40°) in relation to the seedlings that were not generated by seed priming (NSP-W100-T40°, NSP-W50-T30°, and NSP-W50-T40°). The same was observed in the seedlings that were not subjected to the stresses (SPSi-W100-T30°) (Figure 1C,D).

Figure 2A shows that SP with SiMPs under RL irradiation increased the cell membrane integrity in seedlings of *M. oleifera* subjected to thermal (SPSi-W100-T40°) and water stresses (SPSi-W50-T30°) either in isolation or combined (SPSi-W50-T40°) by reducing electrolyte leakage (EL). This EL reduction was not observed in non-stressed seedlings (SPSi-W100-T30°). When SP was not applied, higher EL values were recorded in the seedlings subjected to thermal (NSP-W100-T40°) and water stresses (NSP-W50-T30°) in isolation.

Figure 2B shows that SP decreased the total soluble sugar content in the roots (TSS-R) of seedlings subjected to water stress (SPSi-W50-T30°) and in the roots of non-stressed seedlings (SPSi-W100-T30°) in relation to those in which SP was not applied (NSP-W50-T30° and NSP-W100-T30°). When SP was not applied, the seedlings subjected to water stress in isolation (NSP-W50-T30°) showed higher TSS-R accumulation.

Figure 2C shows that SP increased the activity of the enzyme superoxide dismutase in the roots (SOD-R) of seedlings subjected to water stress only (SPSi-W50-T30°) in relation to those that did not receive SP (NSP-W50-T30°). When SP was not applied, the seedlings subjected to thermal and water stresses combined (NSP-W50-T40°) and those that were not stressed (NSP-W100-T30°) showed high SOD-R activity. However, SP reduced the enzyme activity in these seedlings (SPSi-W50-T40° and SPSi-W100-T30°). SP also reduced the SOD-R activity in the seedlings only subjected to thermal stress (SPSi-W100-T40°).

Figure 3 shows the differences caused by thermal and water stress on the growth of *M. oleifera* at 35 days after sowing. Under thermal stress (40 °C day/35 °C night), the seedlings showed shorter petioles and smaller leaves with a dark green color. Under water stress (W50), the seedlings showed less height. However, seed priming with SiMPs under red light irradiation reversed the differences caused by the combined stresses.

The means of the scores for each PC, the means comparison test for these scores, and the means and standard error of the original individual variables can be seen in Table 2. These results reinforce the differences caused by the interaction between temperature variations, soil water replenishment levels and seed priming, as seen in Figure 1.

## 3. Discussion

*M. oleifera* is a tropical tree considered tolerant to soil water restriction and high-temperature stress. Although the mechanisms that allow this tolerance are still poorly understood, one of the adaptive metabolic responses when seedlings are exposed to heat stress (35 °C day/18 °C night) is accumulating non-structural carbohydrates, proline, and phenols for osmotic adjustment and reducing water loss to the environment through gas exchange [12].

In our study, in PC_1_, the seedlings that were not subjected to SP, reduction of osmotic adjustment indicators (PRO, TSP-L, and TSS-L), and increased gas exchange (gs, E, and Ci) showed an imbalance of the tolerance mechanism resulting from the severity of the thermal stress imposed (40 °C day/35 °C night). The optimal temperature for photosynthesis generally ranges from 20 to 30 °C, with limited enzyme activity and even denaturation of enzymes of the photosynthetic complex at higher temperatures (above 45 °C, for example) [26]. In fact, the increase in CAT-L activity shows that heat stress triggered photosynthetic limitations responsible for decreasing TDM accumulation, justifying the higher RWC in leaflets, which is also related to the water supply resulting from the catalysis of hydrogen peroxide molecules (H_2_O_2_).

Under water stress conditions, Moringa seedlings show reduced leaf water content, reduced leaf water potential, loss of turgidity, stomatal closure, inhibited photosynthesis, impaired metabolic processes, reduced cell expansion, and affected growth and development [17]. In our research, in PC_2_, physiological disorders were also observed in seedlings subjected to water stress (50% of ETc) since the SOD-L activity increased, possibly to catalyze the dismutation of the superoxide anion (O_2_^●−^) in O_2_ and H_2_O_2_ while decreasing A and TDM accumulation due to the reduction in guard cell turgidity and the consequent reduction in the stomatal opening, resulting in lower cell expansion and seedling growth.

When SPSi was applied with RL irradiation, the effects of the isolated or combined heat and water stresses were mitigated by the increased A, PRO, TSP-L, TSP-R, and TSS-L levels under these stress conditions, in addition to TDM accumulation and reductions in RWC, gs, E, Ci, and CAT-L, CAT-R, and SOD-L activity (Figure 4). This mitigation occurs as the SiMPs cross cell wall barriers and, when in contact with plant cells, are captured by plasmodesmata pathways and translocated by apoplastic and/or symplastic pathways [19,23], reducing lipid peroxidation by decreasing the H_2_O_2_ levels, which restores the redox balance and mitigates oxidative stress [25]. In addition, RL could have potentiated the effect of SiMPs since RL irradiation activates the phyA, phyB, phyC, phyD, and phyE phytochromes responsible for light capture and modulation, gene expression, and the transduction of signals related to growth regulation, plant development, metabolic activities, and responses to heat and water stress [2,27].

EL increased in the leaflets of *M. oleifera* seedlings subjected to heat and water stress when SP was not applied, which was also noted by [28], when they observed that water deficit (18 days without irrigation) induced an EL increase since the accumulation of reactive oxygen species (ROS) caused lipid peroxidation and rupture of the thylakoid membrane, thus limiting photosynthesis. However, in our study, there was a reduction in EL in *M. oleifera* seedlings subjected to thermal and water stress when SP was applied, which could be due to the role of SiMPs in modulating cell wall plasticity, leaf thickness, water-plant ratio, and plant metabolism and defense [29].

The highest TSS-R accumulation in seedlings generated without SP and subjected to water stress was due to osmotic adjustment since, under water restriction conditions, plants prevent water loss through stomatal closure, growth modulation, and proline and total soluble sugar accumulation while the root growth is stimulated, characteristics associated with tolerance to water deficit [30]. However, when SP was applied, the TSS-R of *M. oleifera* seedlings decreased under water stress, indicating that SiMPs can act in the root system of the species as an osmotic adjuster in place of organic osmoregulatory since Si deposition on the cell wall, besides acting as a physical barrier against water loss, also increases the osmotic potential for maintaining the water status [31].

From this perspective, based on the increased SOD-R activity observed in seedlings generated by SP when subjected to water restriction, it can be inferred that *M. oleifera* seedlings activate the antioxidant mechanism of the root system instead of performing osmotic adjustment by organic pathways to overcome the negative effects of water stress. On the other hand, SOD-R decreased when heat stress was imposed, indicating that the tolerance induced by SiMPs in the root system is more efficient under water stress than under heat stress. A SOD-R increase was also observed by [32] in response to water stress during the initial stage of plant development, which was justified by the root being the main organ of initial perception of low soil water availability.

Therefore, as already stressed by [2], the photomorphogenesis of *Moringa oleifera* is mediated by the interaction between light, water, and phytohormones during SP, reinforcing the idea that RL has a synergistic effect on the action of SiMPs since irradiation with RL activates phyB, whose activity reduces the levels of abscisic acid (ABA), a germination repressor, and increases the levels of gibberellic acid (GA), which stimulates germination [33] and could have generated a memory effect when the seeds were dehydrated. This information is essential for sustainably managing agroecosystems cultivated with *M. oleifera*, especially in semi-arid regions where thermal and water stresses are frequent.

## 4. Materials and Methods

### 4.1. Experimental Design

The experiment was set up in a completely randomized design arranged as a 2 × 2 × 2 factorial, with 5 replicates, totaling 40 experimental units. The seed priming factor (SP) consisted of a control (NSP—no seed priming, n = 20), in which the seeds were not treated, and seed priming with glass microparticles used as a Si source (SPSi, n = 20) under irradiation with monochromatic red light (RL) with an emission of 184 lumens m^−2^ and wavelengths from 600 to 680 nm. The soil water replenishment factor (SWR) consisted of two levels (W50—50% and W100—100% of crop evapotranspiration). The temperature change factor (TC) consisted of two temperature variations (TC30°—30 °C day/25 °C night and TC40°—40 °C day/35 °C night, with a 12-h photoperiod).

### 4.2. Seed Priming Application

Seed priming was performed at the Laboratory of Plant Physiology of the Agricultural Engineering Academic Unit (UAEA) of the Center of Technology and Natural Resources (CTRN) of the Federal University of Campina Grande (UFCG), located in the municipality of Campina Grande, Paraíba, Brazil. The location has a semi-arid climate, a mean temperature of 25 °C, and a relative air humidity ranging from 72 to 91% [34].

A biochemical oxygen demand (B.O.D.) germination chamber was adapted to provide monochromatic red light using panels with RGB (red, green, and blue) LED (light emitting diode) lamps with an emission of 184 lumens m^−2^. Then, the seeds of *Moringa oleifera* Lam. obtained from three parent plants that were three years old located in Catolé do Rocha, Paraíba, Brazil, were disinfected with sodium hypochlorite (1%) for three minutes [35] at 25 °C and under RL irradiation.

Next, the seeds were placed in Gerbox^®^ boxes measuring 11 × 11 × 3.5 cm in length, width, and height, respectively, for soaking in a solution containing residual glass microparticles used as a silicon source (600 mg L^−1^ of SiMPs). For that purpose, residual amber glass resulting from beverage bottles was collected at the municipal landfill of Toledo, Paraná, Brazil and contains in its composition more than 75% of SiO_2_ and other elements such as Fe, S, Na, K, Ca and Al. The clean and dry bottles were manually ground and sieved through a 400 Tyler mesh fabric to obtain an upper powder particle size below 38 μm. Next, the boxes were placed in the germination chamber at 25 °C for 24 h. Subsequently, the seeds were transferred to open Gerbox^®^ boxes with two layers of dry germitex paper and dried for 72 h at the same light and temperature conditions used during inhibition.

### 4.3. Seedling Formation, Temperature Variations, and Soil Water Replenishment

The control seeds (NSP—no seed priming) and those obtained by seed priming (SPSi) were sown in pots with a volumetric capacity of 0.3 dm^3^ filled with a substrate composed of sandy soil and earthworm humus at a ratio of 3:1 and with moisture close to field capacity. The pots containing the seeds were then transferred to a Phytotron growth chamber (Weiss Technik, Technal) located at the Experimental Unit belonging to the State University of Paraíba (UEPB), Campina Grande, Paraíba, Brazil.

Given the availability of a single phytotron chamber and in order to meet the temperature variations, the experiment was conducted in two stages, the first with the condition of 30 °C day/25° night and an air relative humidity (RH) ranging from 50 to 60% (T30°), and the second with 40 °C day/35 °C night and an RH ranging from 40 to 50%.

Substrate moisture management was performed in a daily irrigation shift using the weighing method [36], according to which the water lost by crop evapotranspiration (ETc) is replenished. The imposition of water stress caused by the replenishment of 50% ETc (W50) was carried out 18 days after sowing (DAS), while the other plots without water stress were irrigated with 100% ETc (W100).

### 4.4. Variables Evaluated

The gas exchange variables were evaluated 30 days after sowing (DAS), whereas cell membrane integrity and the leaf water status, osmotic adjustment indicators, antioxidant mechanism activity, and total dry matter accumulation were evaluated 35 days after sowing. The analyses were performed at the Laboratory of Ecophysiology of Cultivated Plants (ECOLAB) of UEPB, located in the Integrated Research Complex Três Marias.

#### 4.4.1. Exchange Evaluations

Gas exchange evaluation was performed in three fully expanded leaflets counted from the base of the seedling to its apex, from 8:00 a.m. to 9:00 a.m., using an infrared gas analyzer—IRGA (Infra-red Gas Analyzer)—GFS 3000 FL with a CO_2_ concentration of 400 ppm and an artificial light source to provide a photon flux of 1000 µmol m^−2^ s^−1^. The following variables were measured: net photosynthetic rate (A, µmol of CO_2_ m^−2^ s^−1^), stomatal conductance (gs, mol of H_2_O m^−2^ s^−1^), transpiration (E, mmol of H_2_O m^−2^ s^−1^), and internal CO_2_ concentration (Ci, µmol m^−2^ s^−1^).

#### 4.4.2. Cell Membrane Integrity and Leaf Water Status

The methodology described in [37] (with adaptations) was used to analyzed the intracellular electrolyte leakage (EL, %) and the relative water content (RWC, %), for which 12 leaf discs measuring 113 mm^2^, 6 for EL and 6 for RWC, were collected 35 DAS using a copper pourer.

The EL was determined by washing the leaf discs three times in deionized water to remove the solutes released during cutting. Then, the material was placed in Petri dishes containing 6 mL of deionized water, after which the plates were stored at 25 °C for two hours. After incubation, the electrical conductivity in the medium (ECi) was determined using a portable conductivity meter (WATERPROOF). Then, the samples were subjected to a temperature of 80 °C for 90 min, after which the conductivity was again measured (ECf), and the leakage of electrolytes was quantified using Equation (1).
(1)$$\mathrm{EL}=\left(\frac{{\mathrm{EC}}_{i}}{{\mathrm{EC}}_{f}}\right)*100\%\;,$$
where EL is cell leakage, EC_i_ is the initial electrical conductivity of the medium (dS m^−1^), and EC_f_ is the final electrical conductivity of the medium.

The RWC was determined by weighing the leaf disks to determine the fresh mass (FMD), after which the material was immediately placed in Petri dishes containing 6 mL of deionized water. Then, the dishes were placed in a B.O.D. incubator at 25 °C and 202 lumens m^−2^. After four hours of exposure, the disks were dried with filter paper and weighed to obtain the turgid disk mass (TMD). Subsequently, the plant material was packed in paper bags and transferred to a forced-air oven at 60 °C for 48 h. Finally, the material was weighed to determine the dry disk mass (DMD), and the RWC was quantified using Equation (2).
(2)$$\mathrm{RWC}=\left(\frac{\mathrm{FMD}-\mathrm{DMD}}{\mathrm{TMD}\;-\mathrm{DMD}}\right)*100\%\;,$$
where RWC is the relative water content, FMD is the fresh mass, MSD is the dry mass, and DMD is the turgid disk mass.

#### 4.4.3. Indicators of Osmotic Adjustment

The proline content (PRO, µmol g^−1^ FM—fresh matter) was determined by the colorimetric method described by [38] and modified in [39]. Initially, 250 mg of fresh leaf tissue was weighed and macerated in 5 mL of 3% sulfosalicylic acid, followed by centrifugation at 2000 rpm for 10 min. Then, the supernatant was removed and stored in tubes with a capacity of 2.5 mL for later determination of the PRO concentration.

The extraction of total soluble proteins (TSP, mg g^−1^ FM) in the leaflets (TSP-L) and roots (TSP-R) was performed using 200 mg of fresh leaflet and root mass, respectively. The plant material was macerated and then received 3.0 mL of potassium phosphate buffer (100 mM, pH 7.0 + EDTA 1 mM), after which the material was stored in Eppendorf tubes for subsequent centrifugation (5000× *g*) for 10 min in a refrigerated centrifuge (4 °C). After extraction, the TSP concentration was determined according to the methodology proposed by [40].

The extract used to measure the total soluble sugars (TSS, mg g^−1^ FM) in the leaflets (TSS-L) and roots (TSS-R) was obtained from 200 mg of fresh leaflet mass and 100 mg of fresh root mass. Initially, these samples were macerated in 2 mL of 80% ethanol (*v*/*v*). Then, the extract was added to Eppendorf tubes with a capacity of 2 mL and taken to a water bath (60 °C) for 30 min, after which the tubes were transferred to a centrifuge (2000× *g*) to obtain and collect the supernatant. After removing the supernatant, another 2 mL of ethanol (80%) was added to the same tubes for a new extraction, followed by heating in a water bath and subsequent transfer to the centrifuge. The supernatants resulting from the two washes were mixed in Falcon tubes and stored in Eppendorf tubes, totaling 4 mL of extract per sample. The concentrations of TSS-L and TSS-R were determined by the “phenol-sulfuric” method described by [41].

#### 4.4.4. Activity of the Antioxidant Mechanism

The antioxidant mechanism activity was evaluated by measuring the enzyme activities of superoxide dismutase (SOD, µmol min^−1^ mg^−1^ TSP) and catalase (CAT, µmol H_2_O_2_ min^−1^ mg^−1^ TSP) in the leaflets (SOD-L and CAT-L) and roots (SOD-R and CAT-R) was determined using the enzyme extract obtained by the same procedure described TPS.

In the determination of SOD activity, the reaction mixture consisting of 0.3 mL of 130 μM methionine, 0.1 mL of 2250 μM p-nitro tetrazolium blue (NBT), 0.1 mL of 3 μM EDTA, 0.2 mL of riboflavin, 0.75 mL of deionized water, and 1.5 mL of 50 mM sodium phosphate buffer at pH 7.8 received 100 μL of the crude enzyme extract. Then, the absorbance was determined at 560 nm, which was subtracted from the absorbance reading of the reaction mixture without the enzyme extract. Under these conditions, one unit of SOD corresponded to the amount of enzyme required to inhibit the photoreduction of NBT by 50% [42].

CAT activity was determined by adding 100 μL of the crude enzymatic extract to 2.9 mL of the reaction medium consisting of 500 μL of 59 mM hydrogen peroxide (H_2_O_2_), 1.5 mL of 0.05 M potassium phosphate buffer at pH 7.0, and 400 μL of deionized water at 30 °C [43]. Enzyme activity was determined by the reduction in absorbance at 240 nm.

#### 4.4.5. Total Dry Matter Accumulation

The total dry matter accumulation (TDM, g) was quantified in the seedlings by sectioning the plants into leaflets, stem, branches, and roots. Then, the plant material was packed in properly identified paper bags and dried to a constant weight in a forced-air oven at 70 °C. After drying, the plant material was weighed on an analytical balance accurate to 0.0001 g, and the TDM was determined by the sum of the dry matter of leaflets, stem, branches, and roots.

### 4.5. Statistical Analysis

The data on the response variables were subjected to the Shapiro–Wilk normality test [44]. Once the assumptions of normality were met, the data on each variable were standardized to obtain the *Z* variable with a null mean ($\bar{X}$= 0.0) and the unit variance (*s*^2^ = 1.0) according to Equation (3).
(3)$$Z=\frac{X-\bar{X}}{s^{2}}\;,$$
where: *X* corresponds to each observation of the data set of the variable, $\bar{X}$ is the mean, and *s*^2^ is the variance of the data set.

The transformed data were subjected to the exploratory procedure of principal component analysis (PCA). The choice of principal components (PCs) was based on eigenvalues higher than one (λ > 1.0) according to the criterion proposed in [45], which explained a percentage of the total variance higher than 10% [2,46]. The original data referring to each PC were subjected to multivariate analysis of variance (MANOVA) by Hotelling’s T-squared test.

The variables not associated with any PC were removed from the PCA and subjected to univariate analysis of variance (ANOVA) by the F-test at 95% of confidence [47]. These analyses were performed using the software Statistica v. 7.0 [48].

## 5. Conclusions

Seed priming with residual glass microparticles used as a silicon source under red light irradiation mitigated the effects of thermal and water stress in seedlings of Moringa oleifera through the homeostasis of gas exchange, leaf water status, osmotic adjustment, and the plant antioxidant mechanism. Additionally, it was shown that silicon can be provided through an accessible and residual glass source on a micrometer scale, which can be achieved through traditional grinding and sieving processes.

## Figures and Tables

**Figure 1 plants-11-02510-f001:**
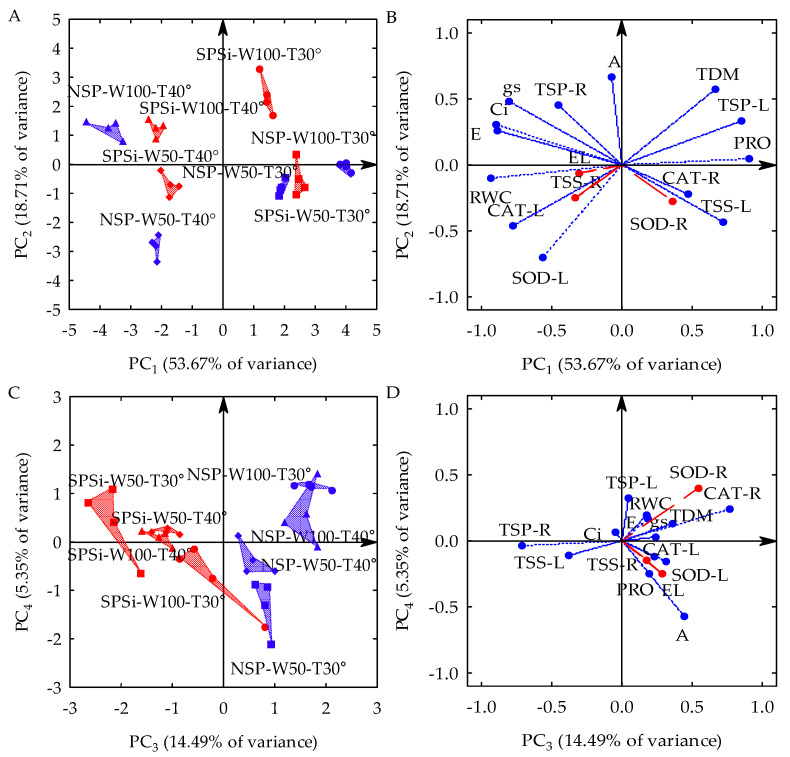
Two-dimensional projection of factorial scores (**A**,**C**) and variables (**B**,**D**) in the first four principal components (PCs 1, 2, 3, and 4) for the interaction between soil water replenishment levels (W50 and W100) and temperatures (T30° and T40°) in *Moringa oleifera*. PC, Principal Component; NSP, no seed priming; SPSi, seed priming with SiMPs; W100, no water stress; W50, water stress; T30°, no thermal stress; T40°, thermal stress; EL, electrolyte leakage; RWC, relative water content; TDM, total dry matter; A, net photosynthetic rate; gs, stomatal conductance; E, transpiration; Ci, internal CO_2_ concentration; PRO, proline; TSP-L, total soluble proteins in leaves; TSP-R, total soluble proteins in roots; TSS-L, total soluble sugars in leaves; TSS-R, total soluble sugars in roots; SOD-L, superoxide dismutase in leaves; SOD-R, superoxide dismutase in roots; CAT-L, catalase in leaves; CAT-R, catalase in roots; ●, NSP-W100-T30°; ▲, NSP-W100-T40°; ■, NSP-W50-T30°; ♦, NSP-W50-T40°; ●, SPSi-W100-T30°; ▲, SPSi-W100-T40°; ■, SPSi-W50-T30°; ♦, SPSi-W50-T40°.

**Figure 2 plants-11-02510-f002:**
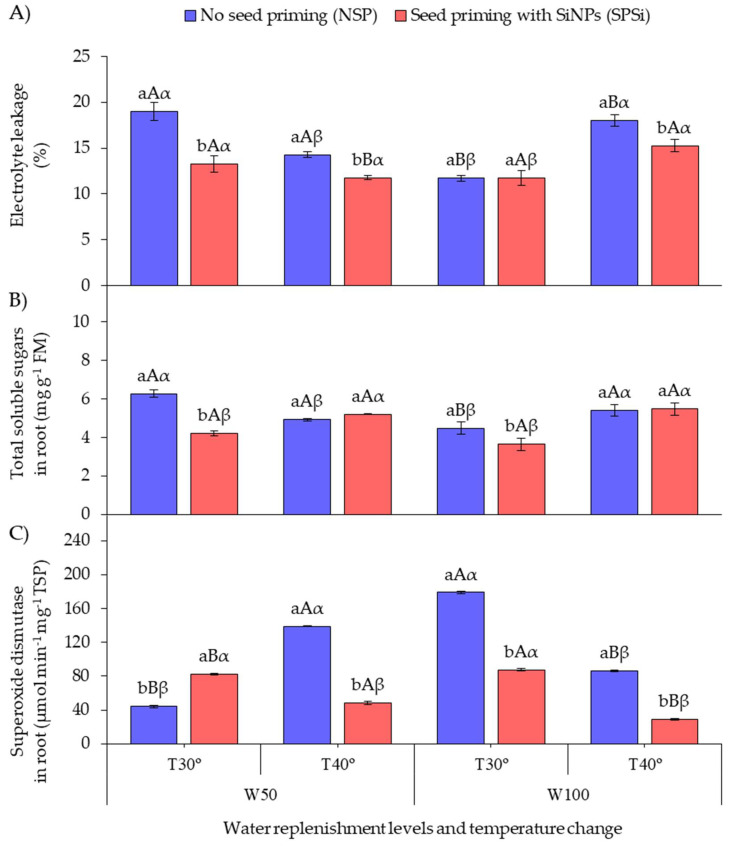
Electrolyte leakage (**A**), total soluble sugars in roots (**B**), and superoxide dismutase activity in roots (**C**) as a function of soil water replenishment levels and temperature change in *Moringa oleifera*. Means followed by the same lowercase letters (a, and b) for seed priming, uppercase letters (A, and B) for soil water replenishment, and Greek letters (α, and β) for temperature change do not differ (*p* > 0.05) by Student’s *t*-test. FM, fresh mass; TSP, total soluble proteins.

**Figure 3 plants-11-02510-f003:**
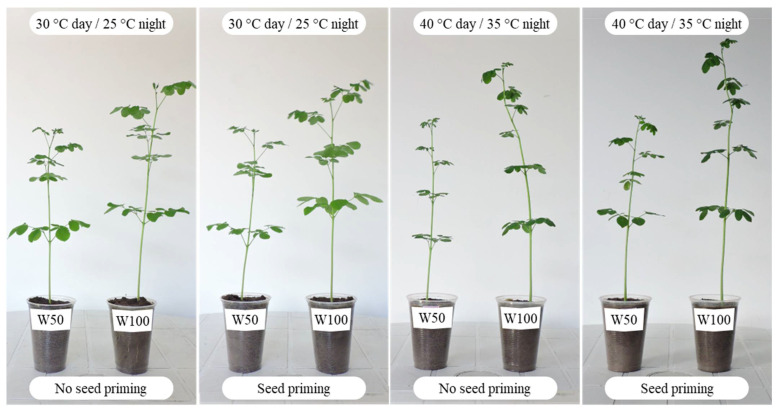
Seedlings of *Moringa oleifera* subjected to seed priming, soil water replenishment, and temperature change. W50, water stress; W100, no water stress.

**Figure 4 plants-11-02510-f004:**
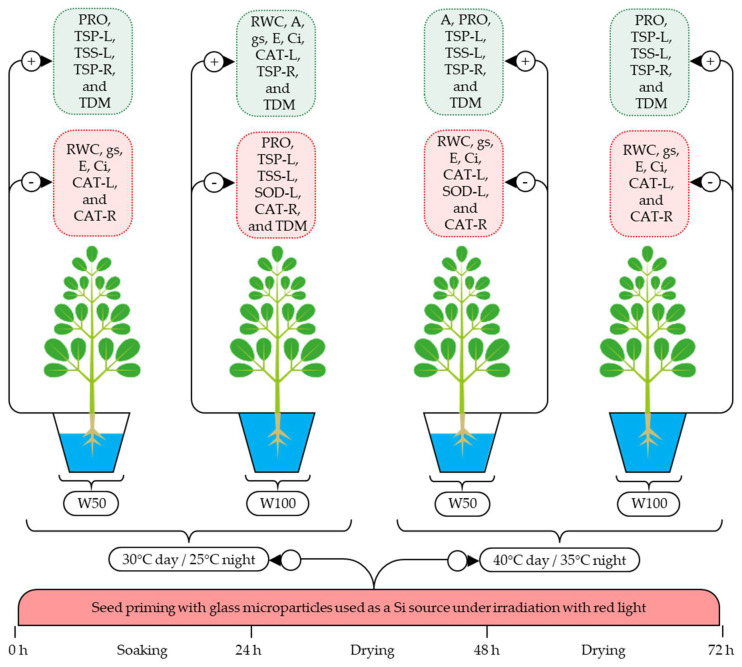
Effect of seed priming with SiMPs under red light irradiation on seedlings of *Moringa oleifera* subjected to thermal and water stress. RWC, relative water content; TDM, total dry matter; A, net photosynthetic rate; gs, stomatal conductance; E, transpiration; Ci, internal CO_2_ concentration; PRO, proline; TSP-L, total soluble proteins in leaves; TSP-R, total soluble proteins in roots; TSS-L, total soluble sugars in leaves; SOD-L, superoxide dismutase in leaves; CAT-L, catalase in leaves; CAT-R, catalase in roots; W50, water stress; W100, no water stress.

**Table 1 plants-11-02510-t001:** Correlation between the original variables and principal components, eigenvalues, explained and cumulative variance, and probability of significance of the hypothesis test in the interaction between the first three principal components (PCs 1, 2, and 3) and seed priming, soil water replenishment levels, and temperature change in *Moringa oleifera* seedlings.

EV—Evaluated Variables	PC—Principal Components		
	PC_1_	PC_2_	PC_3_
EL—Electrolyte leakage	−0.30	−0.06	0.28
RWC—Relative water content	−0.94 *	−0.10	0.18
A—Net photosynthetic rate	−0.07	0.67 *	0.45
gs—Stomatal conductance	−0.80 *	0.48	0.24
E—Transpiration	−0.89 *	0.26	0.17
Ci—Internal CO_2_ concentration	−0.89 *	0.30	−0.05
PRO—Proline in leaves	0.91 *	0.05	0.19
TSP-L—Total soluble proteins in leaves	0.85 *	0.34	0.05
TSP-R—Total soluble proteins in roots	−0.46	0.45	−0.71 *
TSS-L—Total soluble sugars in leaves	0.73 *	−0.43	−0.38
TSS-R—Total soluble sugars in roots	−0.33	−0.25	0.17
SOD-L—Superoxide dismutase in leaves	−0.56	−0.70 *	0.32
SOD-R—Superoxide dismutase in roots	0.36	−0.28	0.54
CAT-L—Catalase in leaves	−0.77 *	−0.46	0.23
CAT-R—Catalase in roots	0.47	−0.22	0.77 *
TDM—Total dry matter	0.66 *	0.57 *	0.36
λ—Eigenvalues	6.98	2.43	1.88
s^2^ (%)—Explained variance	53.67	18.71	14.49
s^2^ (%)—Cumulative variance	53.67	72.38	86.87
**MANOVA—Multivariate analysis of variance**	**Significance probability (*p*-value)**		
Hotelling’s T-squared test for seed priming—SP	<0.01	<0.01	<0.01
Hotelling’s T-squared test for soil water replenishment—SWR	<0.01	<0.01	<0.01
Hotelling’s T-squared test for temperature change—TC	<0.01	<0.01	<0.01
Hotelling’s T-squared test for the SP × SWR interaction	<0.01	<0.01	<0.01
Hotelling’s T-squared test for the SP × TC interaction	<0.01	<0.01	<0.01
Hotelling’s T-squared test for the SWR × TC interaction	<0.01	<0.01	<0.01
Hotelling’s T-squared test for the SP × SWR × TC interaction	<0.01	<0.01	<0.01

Variables considered in PC formation (*).

**Table 2 plants-11-02510-t002:** Means of factor scores for each component and means ± standard error of the original variables evaluated as a function of the interaction between seed priming, soil water replacement levels, and temperature change in *Moringa oleifera* seedlings.

Principal Components	No Seed Priming				Seed Priming with SiMPs			
	W50		W100		W50		W100	
	T30°	T40°	T30°	T40°	T30°	T40°	T30°	T40°
	Factor Score Means for Each Component							
PC_1_	0.72bBα	−0.82bAβ	1.51aAα	−1.41bBβ	0.93aAα	−0.65aAβ	0.53bBα	−0.82aBβ
PC_2_	−0.52aBα	−1.80bBβ	−0.04bAβ	0.80aAα	−0.31aBα	−0.45aBα	1.52aAα	0.81aAβ
PC_3_	0.58aBα	0.41aBα	1.25aAα	1.17aAα	−1.56bBβ	−0.81bAα	−0.14bAα	−0.91bAβ
**Variables**	**Means ± Standard error of the original variables**							
EL (%)	19.00 ± 0.32	14.27 ± 1.02	11.71 ± 0.66	18.00 ± 0.32	13.27 ± 0.19	11.78 ± 0.92	11.73 ± 0.66	15.27 ± 0.80
RWC (%)	83.72 ± 0.79	99.75 ± 0.37	79.81 ± 0.33	111.88 ± 1.00	79.65 ± 0.33	101.98 ± 0.95	79.62 ± 0.67	95.52 ± 0.86
TDM (g)	0.56 ± 0.01	0.31 ± 0.00	1.03 ± 0.01	0.53 ± 0.01	0.52 ± 0.01	0.31 ± 0.01	0.95 ± 0.03	0.54 ± 0.02
A (µmol of CO_2_ m^−2^ s^−1^)	3.09 ± 0.09	2.67 ± 0.07	2.79 ± 0.03	3.13 ± 0.06	2.53 ± 0.13	2.68 ± 0.02	3.30 ± 0.14	2.91 ± 0.03
gs (mol of H_2_O m^−2^ s^−1^)	16.00 ± 0.45	20.83 ± 0.05	14.53 ± 0.34	33.97 ± 0.16	14.70 ± 0.46	19.90 ± 0.71	22.93 ± 0.61	24.93 ± 1.09
E (mmol of H_2_O m^−2^ s^−1^)	0.49 ± 0.02	0.67 ± 0.01	0.43 ± 0.02	1.10 ± 0.09	0.45 ± 0.00	0.75 ± 0.02	0.57 ± 0.00	0.81 ± 0.01
Ci (µmol m^−2^ s^−1^)	86.37 ± 1.29	181.33 ± 2.68	78.53 ± 0.78	233.07 ± 3.47	119.90 ± 0.18	152.03 ± 1.42	167.17 ± 0.78	199.67 ± 1.56
PRO (µmol g^−1^ FM)	796.05 ± 2.16	60.68 ± 0.83	650.88 ± 10.7	27.25 ± 1.04	475.88 ± 2.06	44.08 ± 0.46	521.93 ± 0.77	28.15 ± 0.35
TSP-L (mg g^−1^ FM)	3.28 ± 0.04	1.08 ± 0.03	8.85 ± 0.03	2.78 ± 0.06	7.04 ± 0.03	1.61 ± 0.03	6.42 ± 0.03	2.61 ± 0.03
TSP-R (mg g^−1^ FM)	2.38 ± 0.12	1.94 ± 0.05	0.92 ± 0.05	3.50 ± 0.16	4.02 ± 0.16	4.58 ± 0.03	3.89 ± 0.14	6.58 ± 0.11
TSS-L (mg g^−1^ FM)	6.41 ± 0.26	6.17 ± 0.14	6.31 ± 0.15	3.83 ± 0.24	8.05 ± 0.13	5.30 ± 0.27	5.76 ± 0.23	4.75 ± 0.06
TSS-R (mg g^−1^ FM)	6.28 ± 0.06	4.93 ± 0.20	4.48 ± 0.30	5.39 ± 0.33	4.23 ± 0.03	5.21 ± 0.13	3.64 ± 0.32	5.47 ± 0.31
SOD-L (µmol min^−1^ mg^−1^ TSP)	69.10 ± 1.23	180.92 ± 5.13	32.55 ± 0.24	89.75 ± 1.57	22.52 ± 1.07	83.62 ± 0.48	21.76 ± 0.42	32.23 ± 0.22
SOD-R (µmol min^−1^ mg^−1^ TSP)	44.26 ± 0.60	138.98 ± 1.54	178.90 ± 0.89	86.46 ± 1.07	82.21 ± 1.57	48.22 ± 1.28	87.32 ± 1.03	29.05 ± 1.32
CAT-L (µmol H_2_O_2_ min^−1^ mg^−1^ TSP)	7.47 ± 0.19	16.47 ± 0.53	3.25 ± 0.15	11.15 ± 0.68	2.41 ± 0.14	7.41 ± 0.68	1.98 ± 0.13	10.68 ± 0.47
CAT-R (µmol H_2_O_2_ min^−1^ mg^−1^ TSP)	26.56 ± 1.42	14.81 ± 0.90	42.60 ± 1.29	18.67 ± 0.58	7.66 ± 0.39	5.91 ± 0.13	4.74 ± 0.55	4.40 ± 0.31

PC, Principal component; W50 and W100, soil water replenishment levels; T30° and T40°, temperature change; EL, electrolyte leakage; RWC, relative water content; TDM, total dry matter; A, net photosynthetic rate; gs, stomatal conductance; E, transpiration; Ci, internal CO_2_ concentration; PRO, proline; TSP-L, total soluble proteins in leaf; TSP-R, total soluble proteins in root; TSS-L, total soluble sugars in leaf; TSS-R, total soluble sugars in root; SOD-L, superoxide dismutase in leaf; SOD-R, superoxide dismutase in root; CAT-L, catalase in leaf; CAT-R, catalase in root. Lowercase letters compare seed priming, uppercase letters compare soil water replenishment levels, and Greek letters compare temperature variations. Means of factor scores for each component followed by the same lowercase letters (a, and b) for seed priming, uppercase letters (A, and B) for soil water replenishment, and Greek letters (α, and β) for temperature change in the same row do not differ (*p* > 0.05) by Student’s t-test.

## Data Availability

Not applicable.

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
