# Peer review of "Seed Priming with Glass Waste Microparticles and Red Light Irradiation Mitigates Thermal and Water Stresses in Seedlings of Moringa oleifera"

_plants, 2022, doi:10.3390/plants11192510_

Round 1

Reviewer 1 Report

This work investigated the influence of seed priming with glass waste microparticles, as silicon source, and of red light irradiation to mitigate abiotic stresses in Moringa oleifera seedlings.
The topic is interesting and well within the aims of the Journal. In general, it is quite well written, but I found all the different abbreviations inserted in the text too complex and difficult to follow so, where it is possible, I suggest to maintain the extended names.
Yet, the manuscript needs minor changes and implementations before it can be fully considered for publication in this Journal.
My suggestions are listed below one by one:

TITLE I suggest to simplify and shorten the title, such as the following: “Seed Priming with Glass Waste Microparticles and Red Light Irradiation Mitigates Abiotic Stresses in Moringa oleifera Seedlings”.

INTRODUCTION I suggest to spend some words to explain how SiMPs acting in the mitigation of abiotic stresses. Line 53 Moringaceae should be written in Italics.

MATERIALS AND METHODS In the experimental design is not clear how many seeds were used for the two different experiments (SP and NSP) and how many seedlings were tested in the different trials. Moreover, I have a curiosity about the percentage of germination, it has been the same in SP and NSP?

Author Response

Comment 1: TITLE I suggest to simplify and shorten the title, such as the following: “Seed Priming with Glass Waste Microparticles and Red Light Irradiation Mitigates Abiotic Stresses in Moringa oleifera Seedlings”.

Authors' response 1: The correction has been made, the title has been modified.

Comment 2: INTRODUCTION I suggest to spend some words to explain how SiMPs acting in the mitigation of abiotic stresses. Line 53 Moringaceae should be written in Italics.

Authors' response 2: The correction was made, a brief explanation of how SiMPs act in mitigating abiotic stresses was inserted.

Comment 3: MATERIALS AND METHODS In the experimental design is not clear how many seeds were used for the two different experiments (SP and NSP) and how many seedlings were tested in the different trials. Moreover, I have a curiosity about the percentage of germination, it has been the same in SP and NSP?

Authors' response 3: The correction was performed, information was entered about how many seeds were used in the experiment and how many seedlings were tested.

Reviewer 2 Report

The manuscript used glass waste microparticles as a silicon source to mitigate the effects of thermal and water stress on seedlings of Moringa oleifera seedling. However, the author should provide the characteristic of the material being used in this study, since it is hard to tell what component is functioning in mitigating the effects of thermal and water stress on seedlings.

Line73-74: the author should give a brief overview of the references.

Line82-86: This section should directly describe the research purpose of this paper.

Line104: SOD-F should be revised as SOD-L.

Line106: SOD-R were not associated with any PC, this result was described follow, but here SOD-R was combined with PC3, please check.

Line119-151: Which table or figure is cited to certify the indicator changes (such as RWC, gs, E, Ci, CAT-L, and so on)?

Line190: when did the author take the photo in figure 3?

Line198-199: Please add a description of the results analysis in Table 2.

Line202: In the Variables column of Table 2, please add the unit of each indicator, and it is recommended to add the letter mark of significance test at the back of means ± standard error.

Line317: What is the mean of “3,5”, should it be revised as 3.5 or 35? Please check.

Line369: In equation 1, the “*100” should be revised as “*100%”.

Line369: In equation 2, the “*100” should be revised as “*100%”.

L408: please explain‘Antioxidant Mechanism’

Author Response

Comment 1: “However, the author should provide the characteristic of the material being used in this study, since it is hard to tell what component is functioning in mitigating the effects of thermal and water stress on seedlings”.

Authors' response 1: The correction was made, the characteristic of the material used in this study was inserted.

Comment 2: Line73-74: the author should give a brief overview of the references.

Authors' response 2: The correction has been made, a brief overview of the references has been inserted.

Comment 3: Line82-86: This section should directly describe the research purpose of this paper.

Authors' response 3: The correction was carried out, the objective was described in a separate section.

Comment 4: Line104: SOD-F should be revised as SOD-L.

Authors' response 4: The correction was performed, the acronym SOD-F was adjusted to SOD-L.

Comment 5: Line106: SOD-R were not associated with any PC, this result was described follow, but here SOD-R was combined with PC3, please check.

Authors' response 5: The correction was performed, the writing was verified and adjusted according to the results in Table 1.

Comment 6: Line119-151: Which table or figure is cited to certify the indicator changes (such as RWC, gs, E, Ci, CAT-L, and so on)?

Authors' response 6: The correction was performed, the respective figures were cited according to the presentation of the results.

Comment 7: Line190: when did the author take the photo in figure 3?

Authors' response 7: The correction was made, the date the photos were taken was entered into the results.

Comment 8: Line198-199: Please add a description of the results analysis in Table 2.

Authors' response 8: The correction was made, the description of the analysis of results in Table 2 was inserted.

Comment 9: Line202: In the Variables column of Table 2, please add the unit of each indicator, and it is recommended to add the letter mark of significance test at the back of means ± standard error.

Authors' response 9: The correction was performed, the unit of each variable in Table 2 was inserted. The univariate analysis of variance and the test of means were applied only to the means of the scores of each main component.

Comment 10: Line317: What is the mean of “3,5”, should it be revised as 3.5 or 35? Please check.

Authors' response 10: The correction has been made.

Comment 11: Line369: In equation 1, the “*100” should be revised as “*100%”.

Authors' response 11: The correction has been made.

Comment 12: Line369: In equation 2, the “*100” should be revised as “*100%”.

Authors' response 12: The correction has been made.

Comment 13: L408: please explain‘Antioxidant Mechanism’

Authors' response 13: The correction was made, an explanation was inserted in the text.

Round 2

Reviewer 2 Report

I have no further comment